# CMT: Co-training Mean-Teacher for Unsupervised Domain Adaptation on 3D Object Detection

## ABSTRACT

LiDAR-based 3D detection, as an essential technique in multimedia applications such as augmented reality and autonomous driving, has made great progress in recent years. However, the performance of a well trained 3D detectors is considerably graded when deployed in unseen environments due to the severe domain gap. Traditional unsupervised domain adaptation methods, including co-training and mean-teacher frameworks, do not effectively bridge the domain gap as they struggle with noisy and incomplete pseudo-labels and the inability to capture domain-invariant features. In this work, we introduce a novel Co-training Mean-Teacher (CMT) framework for unsupervised domain adaptation in 3D object detection. Our framework enhances adaptation by leveraging both source and target domain data to construct a hybrid domain that aligns domain-specific features more effectively. We employ hard instance mining to enrich the target domain feature distribution and utilize class-aware contrastive learning to refine feature representations across domains. Additionally, we develop batch adaptive normalization to fine-tune the batch normalization parameters of the teacher model dynamically, promoting more stable and reliable learning. Extensive experiments across various benchmarks, including Waymo, nuScenes and KITTI, demonstrate the superiority of our CMT over the state-of-the-art approaches in different adaptation scenarios.

## CCS CONCEPTS

• **Computing methodologies → Computer vision representations**; **Semi-supervised learning settings**; **Scene understanding**.

## KEYWORDS

Domain Adaptive 3D Detection, Unsupervised Domain Adaptation, 3D Object Detection

## 1 INTRODUCTION

As a cornerstone of 3D scene understanding in multimedia applications and multimodal processing, 3D detection from point clouds has attracted substantial interest. It plays a critical role in intelligent robotics, augmented reality, and autonomous driving [1, 11, 20, 24, 25]. The research primarily aims to detect and localize traffic-related objects within 3D point clouds. With the advent of deep learning technologies, significant advancements have

*ACM MM, 2024, Melbourne, Australia*

© 2024 Copyright held by the owner/author(s). Publication rights licensed to ACM.
ACM ISBN 978-x-xxxx-xxxx-x/YY/MM
https://doi.org/10.1145/nnnnnnn.nnnnnnn

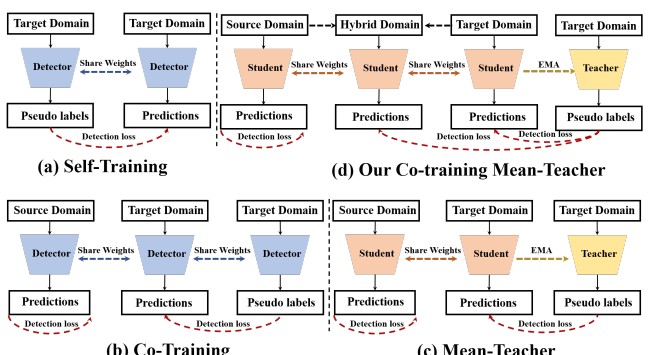

**Figure 1: The comparison of our Co-training Mean-Teacher framework (d) with the 3 previous frameworks (a), (b), (c).**

been made [19, 31, 32, 38], significantly improving the accuracy and efficiency of object detection in complex multimedia environments in recent years, which however requires costly dense annotations of point clouds. Despite significant process, some state-of-the-art 3D detectors still suffer from dramatic performance degradation when training data and test data are from different environments, i.e., domain shift problem [35]. For example, 3D detectors trained on data collected in American cities showed a 45% performance degradation when evaluated on data from European cities [39]. Various factors, such as diverse weather conditions [37], object sizes [35], laser beams [36], lead to substantial discrepancies across different domains, hindering the transferability of existing LiDAR-based 3D detectors. Though collecting more training data from different domains could alleviate this problem, but it might be infeasible due to the multiple real-world scenarios and enormous costs for 3D annotation [22]. Therefore, the research on unsupervised domain adaptation (UDA) for LiDAR-based 3D detection is essential.

Although many works have been proposed to deal with the UDA for image-based detection [3, 6, 9, 15–17, 45, 46], directly applying these methods to 3D point cloud detection task is insufficient for tackling the domain shifts. These approaches mainly concentrate on the gaps of lighting and texture variations, which could not be obtained from point clouds.

Prior works for UDA on 3D object detection typically start from pre-training a model on source labeled data with Random Object Scaling (ROS) [5, 39, 40] and Random Beam Re-Sampling (RBRS) [14, 36] to mitigate domain shifts induced by object sizes and laser beams. Then, the model is further fine-tuned on unlabeled target domain data until convergence is achieved. Specifically, self-training methods [12, 13, 39] iterate between generating pseudo labels and training the model on unlabeled target data as shown in figure. 1(a), which is prone to accumulating errors due to noise in pseudo labels. During the fine-tuning stage, these methods mainly focus

on exploiting the unlabeled information in the target domain. Besides, some works like ST3D++ [40] adopt co-training [21] to jointly optimize source data and pseudo-labeled target data as shown in figure. 1(b). This method effectively corrects inaccurate gradients from noisy pseudo labels. Furthermore, With the teacher model providing supervision to the student outputs of the unlabeled data as shown in figure. 1(c), the mean-teacher framework [23] effectively mitigates errors introduced by pseudo label noise through joint optimization.

However, current mainstream adaptation frameworks face several clear limitations. (1) **Inadequate domain-invariant features:** These frameworks either do not fully utilize the extensively labeled source domain data for effective cross-domain knowledge transfer during the fine-tuning phase, such as with self-training methods, or they simply involve joint training of source and target domain data, as seen in co-training and mean-teacher methods. In these methods, the two independent learning branches frequently suffer from the bias of fully labeled source data, making it challenging for the model to capture domain-invariant features. (2) **Incomplete labels:** The generation of pseudo-labels in the target domain is often incomplete, resulting in a limited distribution of learned target domain foreground features. This situation cause the model to focus on learning simple and similar features while neglecting complex and diverse ones, thereby undermining the model's generalization capabilities and robustness. (3) **Noisy labels:** Traditional frameworks typically rely on inaccurate pseudo-labels for training in the target domain. This reliance leads to an accumulation of incorrect pseudo-labels, forcing the student model to mistakenly adapt its detection outcomes to these noisy labels. Over time, these inaccurate predictions become detrimental learning signals, significantly limiting the model's performance in the target domain.

To address the above issues, we design a novel Co-training Mean-Teacher (CMT) framework for UDA in 3D object detection, as shown in Figure. 1 (d). The key idea of the CMT is to apply **co-training across the source and target domains** while employing the **mean-teacher learning with both the target domain and a hybrid domain** (constructed from source and target domains). This approach is designed to mitigate the impact of source domain data and **facilitate the learning of domain-invariant features**.

Specifically, we first create a hybrid domain using the source domain backgrounds and target domain foregrounds, ensuring that the hybrid domain data integrates knowledge from both domains. The identical foreground in both the hybrid and target domains means they share the same supervision signals. To address the **incomplete label issue**, we introduce a hard instance mining module to extract hard instances from labeled source data and incorporate them into the current target samples, thereby enriching the diversity of the target domain. Consequently, both the hybrid and target domains possess a rich set of foreground features for training the model.

We then feed both the student and teacher models with samples from the hybrid and target domains in mean-teacher learning. To minimize the impact of wrong pseudo-labels and encourage the model to learn domain-invariant features, we develop a class-aware contrastive learning module. This module is based on objects predicted by both teacher and student models, offering more reliable supervision to address the **noisy label issue**. Features from

the same category form a positive sample space, while those from different categories create a negative space, reducing intra-class distances and increasing inter-class distances in the feature space to improve feature discrimination for 3D detection.

During the training, the student model is optimized based on the predictions of the teacher while the weights of the teacher model are updated by taking the exponential moving average of the weights of the student model. however,

Additionally, we develop the framework with batch adaptive normalization to adjust the Batch Normalization (BN) layer parameters of the teacher model with each target training batch, enhancing adaptation to the target domain distribution. In this way, our CMT effectively transfers knowledge across domains, achieving state-of-the-art performance on target data.

To summarize, our contributions are as follows:

- We design a novel Cross-domain Teacher-Student (CMT) framework for unsupervised domain adaptation. By integrating knowledge from both source and target domains, we construct a hybrid domain to facilitate knowledge transfer to the target domain. We develop this architecture using batch adaptive normalization to adjust the Batch Normalization (BN) layer parameters of the teacher model with each target training batch, thereby improving adaptation to the target domain distribution.

- we design a hard instance augmentation method and a background replacement module to create hybrid domain data that combines knowledge from both the source and target domains. We also implement class-aware contrastive learning to keep the class-aware consistency between the hybrid domain and the target domain.

- Experimental results demonstrate our model significantly outperforms the state-of-the-art methods on three widely adopted 3D object detection datasets including NuScenes, KITTI, and Waymo.

## 2 RELATED WORK

### 2.1 LiDAR-based 3D Detection.

LiDAR-based 3D detectors [19, 29, 31, 38] aim to localize and classify 3D objects from point clouds, which can be broadly grouped into three categories: voxel-based methods, point-based methods and point-voxel-based methods. Voxel-based methods [19, 38, 44] typically convert point clouds into regular 3D voxels, subsequently compressing them into a bird's-eye view (BEV) representations. These methods facilitate efficient computation for feature extraction through 2D/3D convolutional neural networks (CNNs). Voxel-Net [44] introduced a voxel-wise encoding layer to extract collective features from voxels, while SECOND [38] adopted sparse 3D convolution for efficient feature learning. PointPillars [19] further proposed a pillar encoding method to voxelise point clouds and convert them into 2D space to improve the efficiency. On the other hand, point-based methods [31, 41] directly process the raw point cloud without voxelization to extract point-wise features through networks of the PointNet series [26, 27]. However, these methods are slower in computation. PointRCNN [31] generated coarse proposals for each point and used the point-level features for further refinement. Point-Voxel-based methods [29, 30] combined voxel

 

representations with point representations from the point cloud. PV-RCNN [29] and PV-RCNN++ [30] proposed to combine voxelization and point-based set abstraction to learn the representative scene features comprehensively. Following previous works, we adopt voxe-based PointPillars [19], SECOND [38] and point-voxel-based PV-RCNN [29] as our base detectors.

## 2.2 Domain Adaptive 2D/3D Object Detection.

A variety of solutions have been proposed in 2D vision tasks including classification [9], detection [8] and segmentation [46]. Inspired by Generative Adversarial Networks [7], adversarial learning methods are leveraged to perform alignment in the feature space. Additionally, self-training [8, 17] methods have been employed to generate pseudo labels for unlabeled target domains. Furthermore, some works adopt the CycleGAN [6] to generate training samples with styles of source and target domains. This methods aids in improving the learning process and adapting more effectively. While extensive researches have been conducted on domain adaptation tasks with 2D image data, only very few approaches have been proposed to address the domain shift in 3D object detection.

For 3D object detection, the Statistical Normalization (SN) [35] method initially highlighted the significance of discrepancies in object size statistics across different domains and mitigating the size gap by normalizing the sizes of objects in the source domain leveraging the statistics of the target domain. However, SN is incompatible with unsupervised adaptation tasks. ST3D [39] introduced the Random Object Scaling (ROS) method to normalize source objects across a broad range, but this method depends on the relative sizes between different domains. To tackle the above issues, Sailor [28] leveraged unsupervised anchor calibration to address object size biases, and SF-UDA$^{3D}$ estimated object size of the target domain through temporal coherency. PLS [4] further leveraged the size of pseudo labels to normalize source objects, achieving better results without relying on target domain statistics.

Other methods focus on generating and enhancing 3D pseudo labels through self-training [39], mean-teacher [23] paradigm, or contrastive learning [21]. ST3D++ [40] refined pseudo labels with a memory bank and introduced Domain Specific Normalization (DSNorm) to better utilize source data during training. UMT [13] applied the uncertainty of predictions to generate more precise and stable pseudo labels. MLC-Net [23] was the first to implement a mean-teacher paradigm, leveraging multi-level consistency to facilitate the 3D cross-domain transfer. DTS [14] developed a density-insensitive learning framework that randomly resamples point cloud beams using the mean-teacher framework to mitigate the domain gap induced by density. 3D-COCO [43] introduced a BEV transformation module to learn cross-domain features through a contrastive learning framework. GPA-3D [21] proposed a geometry-aware prototype alignment, employing a soft contrast loss to bridge the domain gap. Although these methods have achieved impressive performance, they often overlook the impacts of insufficient and inaccurate pseudo-labels in the target domain, and fail to effectively learn domain-invariant features. In contrast, our CMT framework introduces hard instance mining and combines co-training with a mean-teacher framework across hybrid and target domains. This strategy mitigates these issues and further improve performance.

## 3 METHODOLOGY

Unsupervised domain adaption for 3D object detection aims to adapt a detector trained on a source domain $D_S = \{(P_i^s, L_i^s)\}_{i=1}^{n_s}$ to an unlabeled target domain $D_T = \{P_j^t\}_{j=1}^{n_t}$, where $n_s$ and $n_t$ indicate the number of point clouds in the source and target domains respectively. Generally, $P_i^s$ and $L_i^s$ represent the $i$-th source input point cloud and its corresponding label. Each point cloud scene $P_i^s$ has the 3-dim spatial coordinates. $L_i^s$ is in the form of object class $k$ and 3D bounding box parameterized by the center location of the bounding box $(c_x, c_y, c_z)$, the size in each dimension $(l, w, h)$, and the heading angle $\theta$. Similarly, $P_j^t$ denotes the $i$-th unlabeled target point cloud. Note that the superscripts $s$ and $t$ stand for source and target domain respectively.

## 3.1 Co-training Mean-Teacher Architecture

Before transferring knowledge from the source to the target domain, a 3D detector must be pre-trained on the annotated source data $D_S$. This pre-training involves the use of standard augmentation techniques: random object scaling (ROS) [42] and random beam resampling (RBRS) [14]. After the model converges in the pre-training phase, it serves as the baseline model for establishing our co-training mean-teacher framework that facilitates knowledge acquisition from both domains. Our framework consists of two separate models, a non-trainable teacher model and a trainable student model, sharing the same architecture.

During our framework learning, we first establish a co-training step. Following previous works [21, 39], the 3D detector is trained on the labeled source domain $D_S$ by minimizing the detection loss $\mathcal{L}_{det}^s$, which is defined as:

$$\mathcal{L}_{det}^s = \mathcal{L}_{reg}^s + \mathcal{L}_{cls}^s \qquad (1)$$

where $\mathcal{L}_{reg}^s$ and $\mathcal{L}_{cls}^s$ represent the regression and classification losses, respectively. We then use the teacher model to generate pseudo-labels $\hat{L}_i^t$. Specifically, predictions from the teacher model with confidence higher than a threshold $c_{th}$ are selected to generate the pseudo labels as:

$$\hat{L}_i^t = \{\hat{l}_j^t \in \hat{L}_i^{\mathcal{T}} | c_j > c_{th}\}, \qquad (2)$$

where $\hat{l}_j^t$ is the $i$-th predicted bounding box in $\hat{L}_i^{\mathcal{T}}$ and $c_j$ is the confidence of $\hat{l}_j^t$. To enhance the foreground feature distribution of the target domain, we utilize a constructed hard instance bank (Sec. 3.2) on source domain data, and randomly sample the instances in the bank to the target domain. The foregrounds of the target domain now combine the original target domain foregrounds with the enriched foregrounds from the hard instance bank. Simultaneously, the target pseudo labels are updated to include both the pseudo labels from the teacher model and the corresponding foreground labels from the hard instance bank.

Under the co-training paradigm, the student detector uses these target pseudo labels and source labels as supervision for training as follows:

$$\mathcal{L}_{co\_training} = \mathcal{L}_{det}^s + \mathcal{L}_{det}^t, \qquad (3)$$

where $\mathcal{L}_{det}^s$ is the detection loss on target data, same as in Eq. (2).

Subsequently, we construct the hybrid domain (Sec. 3.2) using the background from the source domain and the foreground from

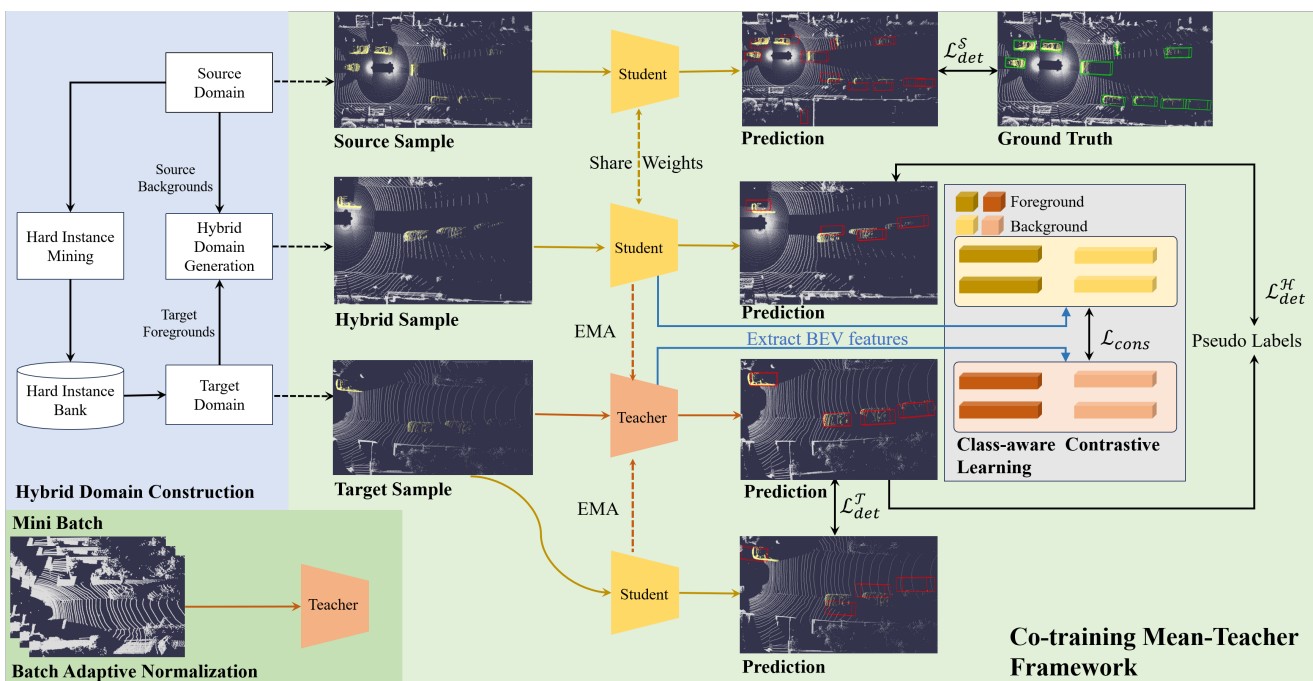

**Figure 2: Our proposed Co-training Mean-teacher (CMT) framework. This figure displays the components of the CMT framework: (1) Hard Instance Mining module extracts hard instances to enhance the diversity of target domain; (2) Hybrid Domain Generation merges source background with target foreground to minimize domain discrepancies; (3) Class-Aware Contrastive Learning improves feature discrimination through a mean-teacher setup that contrasts class features. (4) Batch Adaptive Normalization adjusts the Batch Normalization layer parameters of the teacher model to better adapt to the target domain data.**

the target domain, as shown in Figure. 2 (Left). We then feed both hybrid domain data and target domain data into the student and teacher models for mean-teacher learning, which outputs $\hat{L}_i^{\mathcal{S}}$ and $\hat{L}_i^{\mathcal{T}}$. The student detector is supervised using these target pseudo labels and source labels as follows:

$$\mathcal{L}_{mean-teacher} = \mathcal{L}_{det}^h + \mathcal{L}_{cons}^h \tag{4}$$

where $\mathcal{L}_{det}^h$ is the detection loss on hybrid data as in Eq. (2) and $\mathcal{L}_{cons}^h$ is the consistency loss from class-aware contrastive learning between the teacher and the student models, which will be further elaborated in Section 3.3. The overall loss $\mathcal{L}$ is calculated as:

$$\mathcal{L} = \mathcal{L}_{co-training} + \mathcal{L}_{mean-teacher} \tag{5}$$

To further improve the quality of the pseudo labels, we also deploy the exponential moving average (EMA) technique to update the weight of the teacher detector as:

$$\theta^{'} = (1 - \alpha)\theta^{'} + \alpha\theta \tag{6}$$

where $\alpha$ is a smoothing coefficient hyperparameter. The moving average in Eq. (6) makes $\theta^{\mathcal{T}}$ evolve more smoothly than $\theta^{\mathcal{S}}$. As a result, the teacher can aggregate information after every step and generate stable predictions of the input.

The significant distribution differences between the source and target domains can lead to inconsistencies in Batch Normalization (BN) layer parameters, affecting the quality of pseudo-label generation and the model's performance on the target domain. To address

this issue, we introduce Batch Adaptive Normalization (BAN), a method that dynamically adjusts the BN layer parameters of the teacher model to better adapt to the target domain's data distribution, which will be further elaborated in Section 3.5.

## 3.2 Hybrid Domain Construction

Considering the substantial domain gap between the source and target domains, naive co-training often struggles to capture domain-invariant features and is instead adversely influenced by the distribution of the source domain. To mitigate this issue, we have devised a strategy that utilizes data from both the source and target domains to construct a hybrid domain, facilitating more effective learning within the target domain. We have developed two key modules: Hard Instance Mining and Hybrid Domain Generation to achieve this.

**Hard Instance Mining**. Although high-scoring pseudo labels are more likely to correctly represent the target domain's foreground, this operation might result in a collection of pseudo labels that contain objects with similar patterns. This issue can prevent the model from learning the target domain's diverse distribution. To learn more representative features of the target domain and ensure a gradual and stable adaptation across the significant domain gap, inspired by the ground truth (GT) sampling augmentation proposed by [38], we suggest injecting foreground objects from the source domain into the target domain to enrich its distribution. We cut

out bounding boxes from labeled source domain scenes to create a source instances bank $\mathcal{B}_S = \{b_i^s\}_{i=1}^{n_{gt}}$. $n_{gt}$ is the number of ground truth boxes in all labeled source data and $b_i^s$ denotes the $i$-th ground truth sample. For $b_i^s$, it consists of the corresponding point clouds in labeled data and its ground truth bounding box.

Considering that hard samples often exhibit a more diverse distribution, we propose to categorize sample difficulty based on the number of points within an object. We select hard samples using a quantile threshold $\gamma$ determined by the number of points in current target objects. Specifically, we construct the **hard instance bank** $\mathcal{B}$ by selecting hard samples from $\mathcal{B}_S$ as follows:

$$\mathcal{B} = \{b_i^s \in \mathcal{B}_S \mid |b_i^s| < \gamma\} \tag{7}$$

Here, $|b_i^s|$ represents the number of points within $b_i^s$. During the training stage, we will randomly select ground truth samples from $\mathcal{B}$ and paste them into the current target training frame. The range for $\gamma$ is from 0 to 100, where $\gamma = 0$ indicates that the hard instance bank contains no samples, and $\gamma = 100$ means that the hard instance bank is identical to the source instance bank. We perform collision detection during the insertion process to ensure that the inserted boxes do not overlap with existing objects in the scene. However, since ground truth bounding boxes are unavailable for unlabeled scenes, we rely on prediction results for collision detection in these cases. Finally, we achieve a merged instances that includes both target pseudo instances and source hard instances. This approach ensures a sufficient number of foreground objects in each scene while enhancing the diversity of objects in the target domain.

**Hybrid Domain Generation.** To generate hybrid domain data, we first use the 3D bounding boxes from the source domain to remove all foreground point clouds, thereby obtaining a clean source domain background. Next, we extract the target domain foreground point clouds from the target domain data using pseudo-labels. It is important to note that the target domain foreground includes enriched foreground features provided by the hard instance bank, and the pseudo-labels also include corresponding annotations. Finally, we combine the source domain background point clouds with the target domain foreground point clouds to create the hybrid domain data. By this design, we integrate knowledge from both the source and target domains to construct the hybrid domain. Compared with the original source domain scenarios, this method significantly enhances the adaptability and effectiveness of the domain transfer.

## 3.3 Class-aware Contrastive Learning

In addition to supervision at the final prediction level, we introduce class-aware contrastive learning to better utilize pseudo-labels for feature refinement. Inspired by previous studies [21, 43], Bird's Eye View (BEV) features are more transferable than low-level 3D features since they have grid structures similar to images. Therefore, we first map the pseudo labels $\hat{l}_i^t$ from the teacher model onto the BEV feature map from either the hybrid or target domain to extract BEV features $F_i^{\mathcal{M}} \in \mathbb{R}^{H \times W \times C}$, where $\mathcal{M} \in \{\mathcal{S}, \mathcal{T}\}$ represents the student and teacher models respectively, $H$, $W$, and $C$ denote the height, width, and number of channels of the feature map respectively, as shown in Figure. 2 (right). Drawing inspiration from supervised contrastive learning, we use the teacher's

predicted classes to leverage learning signals from pseudo-labels. The contrastive loss is formulated as follows:

$$\mathcal{L}_{cons} = \frac{\lambda}{N} \sum_{i=1}^{N} \frac{-1}{|\mathcal{P}(i)|} \sum_{p \in \mathcal{P}(i)} log \frac{exp(F_i^{\mathcal{S}} F_p^{\mathcal{T}} / \tau)}{\sum_{j=1}^{N} exp(F_i^{\mathcal{S}} F_j^{\mathcal{T}} / \tau)} \tag{8}$$

where the positive pair set $\mathcal{P}(i) = \{P \mid C_P = C_i, p \in \{1, \cdots, N\}\}$ includes all objects of the same predicted class as object i. The balancing weight $\lambda$ and temperature $\tau$ are hyperparameters.

## 3.4 Batch Adaptive Normalization

Batch Adaptive Normalization (BAN) uses target domain data to update the BN layer parameters of the teacher model within each training batch. Specifically, for the data in each batch, we calculate the batch mean $\mu$ and variance $\sigma^2$ as:

$$\mu = \frac{1}{mp} \sum_{i=1}^{m} \sum_{j=1}^{p} f_{ij}, \quad \sigma^2 = \frac{1}{mp} \sum_{i=1}^{m} \sum_{j=1}^{p} (f_{ij} - \mu)^2, \tag{9}$$

where $m$ is the total number of domain specific samples in a mini-batch, $f$ is the input feature of one feature channel, and $p$ is the number of elements (e.g., pixels or points) in this feature channel. Here, we ignore the channel index $c$ for simplicity, and the above process is performed on each channel separately. Meanwhile, since the transformation parameters $\gamma$ and $\beta$ are domain agnostic and transferable across domains, the two domains shares the learnable scale and shift parameters as:

$$\hat{f}_i = \frac{f_i^d - \mu}{\sigma^2 + \epsilon}, \quad g_i = \gamma \hat{f}_i + \beta, \tag{10}$$

where $\hat{f}$ is the normalized value and $g$ is the transformed feature. During training, the batch mean $\mu$ and variance $\sigma$ are used to update the running statistics at every iteration as Eq. 6. We argue that this modification closes the gap caused by domain mismatch. We empirically demonstrate the effectiveness by comparing the performance under different BN settings in Section 4.3.

## 4 EXPERIMENTS

### 4.1 Experimental Settings

**Datasets.** We conduct experiments on three widely used autonomous driving datasets: KITTI [10], Waymo [33], and nuScenes [2]. The KITTI dataset is collected with 64-beam Velodyne LiDAR. The Waymo dataset is collected with five LiDAR sensors, *i.e.*, one 64-beam LiDAR and four 200-beam LiDARs. The nuScenes dataset is collected with a 32-beam roof LiDAR. These datasets exhibit significant diversities in foreground patterns and LiDAR beams. Following previous works [12, 39, 43], we evaluate our CMT by adapting across domains (Waymo → nuScenes, Waymo → KITTI and nuScenes → KITTI).

**Evaluation metrics.** We follow [35] and adopt the KITTI evaluation metric for evaluating our methods on the commonly used car category (also named vehicle in Waymo). In detail, we follow the official KITTI evaluation metric and use the average precision (AP) for both the 3D IoUs and the bird's eye view (BEV) under an IoU threshold of 0.7 over 40 recall positions. We also adopt the domain adaptation metric ( *i.e.*, Closed Gap) [39] to

**Table 1: Performance comparison of various methods in cross domain adaptation tasks. W, N, and K denote the Waymo, nuScenes, and KITTI datasets, respectively. Methods that require additional information from the target domain are marked with $\dagger$ and labeled as weakly-supervised. "Source Only" indicates that the model trained on the source dataset is directly tested on the target dataset. "Oracle" indicates that the model is trained with labeled target data. We report $AP_{BEV}$ and $AP_{3D}$ over 40 recall positions of the car category at IoU = 0.7.**

| Task | Method | SECOND-IoU | | PV-RCNN | | PointPillars | |
|---|---|---|---|---|---|---|---|
| | | $AP_{BEV}$ / $AP_{3D}$ | Closed Gap | $AP_{BEV}$ / $AP_{3D}$ | Closed Gap | $AP_{BEV}$ / $AP_{3D}$ | Closed Gap |
| W → K | Source Only | 67.64/27.48 | -/- | 61.2/22.0 | -/- | 47.8/11.5 | -/- |
| | SN$^\dagger$ [CVPR'20] | 78.96/59.20 | +72.33%/+69.00% | 79.8/63.6 | +66.9%/+68.7% | 27.4/6.4 | -55.1%/-8.5% |
| | UMT | 77.79/64.56 | +64.86%/+80.66% | -/- | -/- | -/- | -/- |
| | 3D-CoCo [NIPS'21] | -/- | -/- | -/- | -/- | 76.1/42.9 | +76.5%/+52.2% |
| | ST3D [CVPR'21] | 82.19/61.83 | +92.97%/+74.42% | 84.10/64.8 | +82.4%/+70.7% | -/- | -/- |
| | ST3D++ [TPAMI'22] | 80.78/65.64 | +83.96%/+83.01% | -/- | -/- | -/- | -/- |
| | GPA-3D [ICCV'23] | 83.79/70.88 | +103.19%/+94.41% | -/- | -/- | 77.29/50.84 | +79.70%/+65.46% |
| | AttProto [WACV'24] | -/66.86 | -/+85.66% | -/- | -/- | -/- | -/- |
| | Ours | **85.19/72.07** | **+112.14%/+97.00%** | **85.93/74.53** | **+88.96%/+86.83%** | **77.74/54.38** | **+80.92%/+71.35%** |
| | Oracle | 83.29/73.45 | -/- | 89.0/82.5 | -/- | 84.8/71.6 | -/- |
| N → K | Source Only | 51.8/17.9 | -/- | 68.2/37.2 | -/- | 22.8/0.5 | -/- |
| | SN$^\dagger$ [CVPR'20] | 78.96/59.20 | +25.1%/+35.4% | 60.5/49.5 | +36.8%/+27.1% | 39.3/2.0 | +26.6%/+2.1% |
| | UMT | 72.33/47.91 | +65.17%/+53.97% | -/- | -/- | -/- | -/- |
| | 3D-CoCo [NIPS'21] | -/- | -/- | -/- | -/- | 77.0/47.2 | +87.4%/+65.7% |
| | ST3D [CVPR'21] | 75.9/54.1 | +76.6%/+59.5% | 78.4/70.9 | +49.0%/+74.3% | 60.4/11.1 | +60.6%/+14.9% |
| | ST3D++ [TPAMI'22] | 80.5/62.4 | +91.1%/+80.0% | -/- | -/- | -/- | -/- |
| | AttProto [WACV'24] | -/56.51 | -/ +63.08% | -/- | -/- | -/- | -/- |
| | Ours | **82.40/69.98** | **+97.14%/+93.67%** | **84.79/72.55** | **+80.14%/+78.04%** | **80.07/53.86** | **+92.37%/+75.05%** |
| | Oracle | 83.3/73.5 | -/- | 88.9/82.5 | -/- | 84.8/71.6 | -/- |
| W → N | Source Only | 32.9/17.2 | -/- | 34.5/21.5 | -/- | 27.8/12.1 | -/- |
| | SN$^\dagger$ [CVPR'20] | 33.2/18.6 | +1.7%/+7.5% | 34.2/22.3 | -1.5%/+4.8% | 28.3/13.0 | +2.4%/+4.7% |
| | UMT | 35.10/21.05 | +11.54%/+21.61% | -/- | -/- | -/- | -/- |
| | 3D-CoCo [NIPS'21] | -/- | -/- | -/- | -/- | 33.1/20.7 | +25.0%/+44.8% |
| | ST3D [CVPR'21] | 35.9/20.2 | +15.9%/+16.7% | 36.4/23.0 | +10.3%/+8.8% | 30.6/15.6 | +13.2%/+18.2% |
| | ST3D++ [TPAMI'22] | 35.7/20.9 | +14.7%/+20.9% | -/- | -/- | -/- | -/- |
| | GPA-3D [ICCV'23] | 37.25/22.54 | +22.88%/+30.06% | -/- | -/- | 35.47/21.01 | +36.18%/+46.41% |
| | AttProto [WACV'24] | -/20.38 | -/17.97% | -/- | -/- | -/- | -/- |
| | Ours | **41.08/23.98** | **+43.05%/+38.31%** | **41.49/26.30** | **+37.58%/+28.07%** | **38.04/21.09** | **+48.30%/+46.82%** |
| | Oracle | 51.9/34.9 | -/- | 53.1/38.6 | -/- | 49.0/31.3 | -/- |

demonstrate the effectiveness on domain adaption, which is defined as $Closed\ Gap = \frac{AP_{model} - AP_{source}}{AP_{oracle} - AP_{source}} \times 100\%$.

**Implementation details.** We validate the proposed CMT on three base detectors, including SECOND-IoU [38], PV-RCNN[29] and PointPillars [19]. We adopte the training settings of the popular point cloud detection codebase OpenPCDet [34] to pre-train our detectors on the source domain. For the following fine-tuning stage, we use Adam [18] and one cycle scheduler to fine-tune the detectors for 50 epochs. The learning rate is set to $1.5 \times 10^{-3}$. The EMA smoothing coefficient hyperparameter $\alpha$ is set to 0.999. We set the confidence threshold $c_{th}$ of pseudo labels to 0.6 and the quantile threshold $\gamma$ of hard instance bank to 25. The hyperparameters of the contrastive loss $\lambda$ and $\tau$ is set to 0.05 and 0.07, respectively.

**Compared Methods.** As shown in Table 1, CMT is first compared with the Source Only method, which indicates directly evaluating the source domain pre-trained model on the target domain. Next, 7 existing works are included in the comparison, namely,

SN [35], UMT [13], 3D-CoCo [43], ST3D [39], ST3D++ [40], GPA-3D [21], and AttProto [12]. SN utilizes target domain statistical object size as extra information to normalize the foreground objects on the source domain. UMT utilizes a mean-teacher framework and leverages model uncertainty to refine pseudo-labels, employing Monte Carlo dropout for uncertainty estimation. 3D-CoCo focuses on learning object-level transferable features to enhance generalization. ST3D and ST3D++ adopt a memory bank to produce high-quality pseudo-labels. GPA-3D leverages the intrinsic geometric relationship from point cloud objects to reduce the feature discrepancy. AttProto introduces an attentive prototype mechanism using self-attention via transformers to learn distinctive class features, especially effective in handling label noise in source-free unsupervised domain adaptation. Additionally, we also compare CMT with the Oracle, which trains the model on the labeled target data, serving as an upper bound for performance.

Table 2: Main ablation results. BAN represents batch adaptive normalization. HIM represents hard instance augmentation. BRM represents background replacement module. CCL represents class-aware contrastive learning.

| Setting | BAN | HIM | BRM | CCL | $AP_{BEV}$ | $AP_{3D}$ |
|---------|-----|-----|-----|-----|------------|-----------|
| (a) | | | | | 80.77 | 65.12 |
| (b) | ✓ | | | | 83.26 | 67.14 |
| (c) | ✓ | ✓ | | | 83.33 | 69.28 |
| (d) | ✓ | ✓ | ✓ | | 83.48 | 70.08 |
| (e) | ✓ | | ✓ | ✓ | 84.59 | 71.35 |
| (f) | | ✓ | ✓ | ✓ | 82.96 | 67.70 |
| (g) | ✓ | ✓ | ✓ | ✓ | **85.19** | **72.07** |

## 4.2 Comparison with State-of-the-art Methods

**Main Results.** As summarized and reported in Tabel 1, our CMT outperforms all compared methods by large margins on all 3D domain adaptation settings. Especially on the nuScenes → KITTI and Waymo → KITTI tasks using the Second-iou [38] backbone, our CMT reduces the *Closed Gap* in $AP_{3D}$ by around 93% to 97% and in $AP_{BEV}$ by about 97% to 112%, indicating that $AP_{BEV}$ for CMT also surpasses that of the Oracle method. With the 3D detector PV-RCNN [29], our CMT outperforms the previous SOTA ST3D [39], by 9.73% in $AP_{3D}$. Furthermore, in adaptation tasks between the large-scale datasets Waymo and nuScenes, our CMT achieves remarkable improvements, reducing the *Closed Gap* in $AP_{BEV}$ by approximately 41% to 48%. The overall results validate the transferability of our CMT on different unsupervised domain adaptation benchmarks, and its ability to generalize to different detection networks.

## 4.3 Ablation Studies

Following previous works [21, 39, 40], all ablation studies are conducted on Waymo → KITTI, using SECOND-IoU as the base detector.

**Main ablation results.** We assess the effectiveness of each component within the CMT framework, as detailed in Table 2. Baseline (a) utilizes mean-teacher training with labeled source data and pseudo-labels from the target domain. As illustrated in (b), integrating our proposed Batch Adaptive Normalization (BAN) into the pre-training phase of the mean-teacher framework enhances performance by 2.03% in $AP_{3D}$ and 2.49% in $AP_{BEV}$, respectively. Furthermore, the inclusion of Hard Instance Augmentation (HIM), Background Replacement Module (BRM) and Class-aware Contrastive Learning (CCL) significantly boosts the results for (c), (d), and (g), thereby demonstrating the efficacy of each component.

Additionally, we explore the importance of each module by removing them individually. By comparing (f) and (g), we observe substantial declines in $AP_{3D}$ and $AP_{BEV}$ by 4.37% and 2.23%, respectively, highlighting the critical role of dynamically adjusting the statistical distribution of the teacher model. Comparisons between (e) and (g) confirm that hard samples from the source domain effectively enrich the target domain's sample distribution. Similarly, the comparison between (d) and (g) shows that the CCL module largely enhances performance in $AP_{BEV}$.

Table 3: Ablation study of the exponential moving average (EMA) update mechanisms for the BN layer parameters of the teacher model. Disabled: No update to the teacher model's BN parameters. Baseline: Updates the teacher model's BN parameters using the student model's via EMA. Neural Statistics Consistency: Uses the parameters from the student model's BN layers for updating the teacher model during training. Ours: Updates the teacher model's BN parameters with each training batch.

| Method | $AP_{BEV}$ | $AP_{3D}$ |
|--------|------------|-----------|
| Disabled | - | - |
| Baseline | 82.96 | 67.70 |
| Neural Statistics Consistency | 82.90 | 67.92 |
| Ours | **85.19** | **72.07** |

**Analysis of Batch Adaptive Normalization.** The ablation study evaluates the impact of various batch normalization (BN) updating strategies on the performance of a teacher model. Table 3 illustrates the importance of updating the teacher model's BN layer parameters; without these updates, the model's training may collapse. By involving Neural Statistics Consistency, the performance of the model is similar to that achieved using the student model's BN parameters updated via EMA. This circumstance likely stems from inconsistencies between the BN layer parameter updates and other learnable parameters of the teacher model, leading to poorer training outcomes. With our method, the teacher model's BN parameters are dynamically updated using training data, which ensures timely adjustments to the correct feature distribution statistics and achieves optimal performance.

**Sensitivity Analysis of Hard Instance Mining.** As shown in the figure 3, we evaluated the impact of different parameters $\gamma$ on the HIM module. Setting the parameter $\gamma$ to 0 means the HIM module is not used. Setting the parameter $\gamma$ to 5 results in a 1.6% increase in $AP_{BEV}$ and 0.72% in $AP_{3D}$. When set to 25, both $AP_{BEV}$ and $AP_{3D}$ reach their performance peaks. However, setting the parameter too high leads to a slight decline in performance, which we attribute to an excess of simple samples affecting the model's generalization ability.

**Sensitivity Analysis of Class-aware Contrastive Learning.** As shown in the figure 4, we evaluate different $\lambda$ to control the weights of the contrastive loss. When $\lambda$ changed, the overall performance remains relatively stable.

**Analysis of Hybrid Domain Construction.** As shown in the Table 4, we tested various methods for constructing hybrid domains using foregrounds and backgrounds from the source and target domains. By mixing the source domain foreground with both source and target domain backgrounds (configurations (b) and (c)), we observed a remarkable performance decrease compared to the baseline model when the target domain foreground is not used in the hybrid domain. This decline is likely because the ultimate goal is to recognize foregrounds in the target domain, and merely learning from the source domain foreground does not sufficiently enhance model performance. Excessive training on source domain

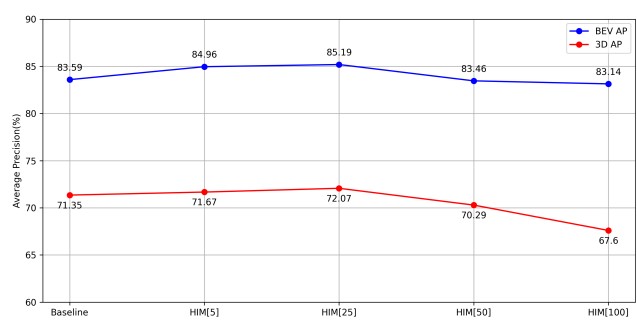

Figure 3: Hyperparameter sensitivity of $\gamma$. Baseline refers to HIM[0]. HIM[$\gamma$] indicates that we use $\gamma$ as the quantile threshold for selecting hard samples.

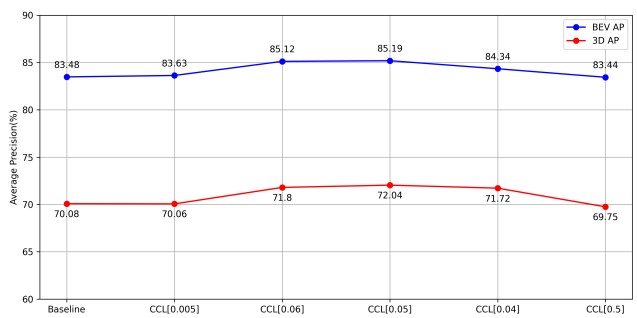

Figure 4: Hyperparameter sensitivity of $\lambda$. Baseline refers to CCL[0]. CCL[$\lambda$] indicates that we use $\lambda$ as the balancing weight of the contrastive loss.

foreground samples can adversely affect the model due to the influence of the source domain distribution, which hinders performance in the target domain.

By mixing the target domain foreground with both source and target domain backgrounds (configurations (d) and (e)), the combination of target domain foreground and source domain background achieved optimal results. We attribute this to the effective use of relevant features from both domains, minimizing domain discrepancy. However, for the combination of target domain foreground and background, the performance is compromised. This is analyzed to be due to the pseudo-labels generated by the teacher model, which may include unrecognized foreground objects within the target background, thereby deteriorating the performance. Additionally, this configuration does not mitigate the influence of the source domain distribution effectively.

**Component Analysis in our framework.** We compare our proposed Cross-domain Teacher-Student (CMT) framework with several established adaptation frameworks, as shown in Table 5. The results clearly demonstrate that CMT significantly outperforms traditional frameworks such as self-training, mean-teacher, and co-training. This improvement emphasizes the effectiveness of our CMT in leveraging cross-domain knowledge.

Furthermore, we conducted a detailed analysis of each component within the CMT training pipeline. Notably, removing self-training from the target domain led to a significant 5.1% drop in $AP_{3D}$, underscoring the importance of this component in adapting

Table 4: Comparative Experiment Results for Hybrid Domain Construction. Here, "Src" denotes Source domain, "Tgt" denotes Target domain, "Fg" denotes Foreground, and "Bg" denotes Background. The combinations (e.g., Src.Fg + Tgt.Bg) represent hybrid domains constructed using foreground (Fg) and background (Bg) components from specified domains.

| Setting | Method | $AP_{BEV}$ | $AP_{3D}$ |
|---|---|---|---|
| (a) | Baseline | 83.33 | 69.28 |
| (b) | Src.Fg + Src.Bg | 82.96 | 65.37 |
| (c) | Src.Fg + Tgt.Bg | 80.89 | 65.00 |
| (d) | Tgt.Fg + Tgt.Bg | 83.33 | 70.19 |
| (e) | Tgt.Fg + Src.Bg | **85.19** | **72.07** |

Table 5: Comparative Experiment Results for Training Frameworks

| Training Framework | $AP_{BEV}$ | $AP_{3D}$ |
|---|---|---|
| Self-Training | 82.25 | 65.95 |
| Mean-Teacher | 80.77 | 65.12 |
| Co-Training | 83.47 | 66.53 |
| Ours w/o Self-T. | 81.91 | 66.99 |
| Ours w/o Mean-T. | 83.68 | 69.76 |
| Ours w/o Co-T. | 83.30 | 69.04 |
| Ours | **85.19** | **72.07** |

the model effectively to the target domain. Direct exposure to target domain data crucially enhances the model's adaptability and accuracy.

Additionally, eliminating mean-teacher and co-training components caused performance degradation, primarily due to the accumulation of incorrect pseudo-labels. These components are crucial for stabilizing the learning process by providing refined and reliable pseudo-labels, which help mitigate error propagation during training. Without these components, the model's generalization under domain shift conditions is severely impaired, highlighting the importance of their inclusion for optimal performance.

## 5 CONCLUSION

In this work, we presented the Co-training Mean-Teacher (CMT) framework for unsupervised domain adaptation in 3D object detection. We address the limitations of current domain adaptation methods by incorporating a hybrid domain that blends source and target domain data, which helps in mitigating the domain discrepancy and enhancing model robustness. Our framework leverages hard instance mining to enrich the target domain with hard examples and employs a class-aware contrastive learning strategy to refine feature representation across domains. By dynamically updating the batch normalization parameters through Batch Adaptive Normalization, our method adapts more effectively to the target domain. Our experimental results demonstrate that CMT outperforms existing state-of-the-art methods on various benchmark datasets. Future research will focus on bridge the gap between synthetic and real-world domains, enhancing the transferability and effectiveness of 3D detection models across simulated and actual environments.

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
