# OpenReview forum: "CMT: Co-training Mean-Teacher for Unsupervised Domain Adaptation on 3D Object Detection"
_acmmm.org/ACMMM/2024/Conference — MM2024 Poster_

### Official Review · Reviewer_gzmV · 2024-05-24

**Rating:** 4
**Confidence:** 1

**Summary:**

This paper proposes a Co-training Mean-Teacher (CMT) framework for unsupervised 3D domain adaptation, which combines Co-training and Mean-Teacher techniques to solve inadequate domain-invariant features, incomplete labels, and noisy labels. Specifically, the authors first construct a hybrid domain by combining the original target domain foregrounds with the source domain backgrounds to guarantee the knowledge integration of both domains. A hard instance mining module is introduced to enrich the diversity of the target domain. Besides, the authors design a class-aware contrastive learning module for learning domain-invariant features to address the noisy label issue. Finally, the extensive experimental results demonstrate the effectiveness of the proposed CMT framework and achieve new SOTA results on nuScenes, KITTI, and Waymo.

**Strengths:**

The paper is well-organized and easy to follow. The solution for unsupervised 3D object detection is technically sound.

**Limitations:**

1. As the era of end-to-end autonomous driving, 3D object detection is no longer an essential component of current AD systems. Therefore, the demand for 3D UDA, a related task of 3D object detection, is also slowly dying off.
2. The pipeline of the proposed CMT is too long and tedious. It is more like an engineering framework than a good research paper.

**Suitability:**

3

---

### Official Review · Reviewer_eJxD · 2024-05-24

**Rating:** 3
**Confidence:** 4

**Summary:**

Inspired by mean-teacher method, this work explores its application in 3D domain adaptive object detection. Specifically, this work presents many module to address the 3D domain adaptation problem. For example, they propose hybrid domain generation module to generat the hybrid samples. Besides, they propose class-aware contrastive learning using the mean-teach setup to learn domain-invariant features. And, the batch-adaptive normalization is proposed to align the BN distribution.

**Strengths:**

- This paper is easy to follow and well organized.
- This paper verifies the effectiveness of their proposed method on many baseline models including SECOND-IoU, PV-RCNN, PointPillar
- The work gives many analyses of each designed module. For example, the sensitivity of class-aware contrastive learning and hard instance mining can be found in experiment section.

**Limitations:**

- In my opinion, the main limitation of this work is that, it appears to be a direct and incremental combination of some trick points. For example, to improve the performance of the target domain, the hybrid domain generation is used to generate hybrid samples, and thus, batch adaptive normalization is used to align the statistical distribution of BN layer, etc. Although the ablation studies are conducted on Table 2 and performance gain can be observed for each proposed module, I still think these performance gains are tricky in domain adaptation. Overall, it is suggested that the authors conduct the in-depth research around one of the points.

- Also, some proposed modules are not novel. For mean-teacher setup, the work [Ref A] also uses the MT to maximally exploit the expertise of the teacher model and boost the model domain adaptability. For batch adaptive normalization, the work [Ref B] also introduces a domain-specific normalization method to reduce the 3D features' gap. Please explain the differences between this work and previous works.

[Ref A] Unbiased mean teacher for cross-domain object detection. CVPR-2021.
[Ref B] Uni3d: A unified baseline for multi-dataset 3d object detection. CVPR-2023

**Suitability:**

3

---

### Official Review · Reviewer_QUvJ · 2024-05-25

**Rating:** 5
**Confidence:** 4

**Summary:**

This study focuses on unsupervised domain adaptation for 3D object detection, aiming to adapt 3D detectors across different 3D scene domains. This paper proposes a co-training mean-teacher framework that incorporates hard instance mining and training with a generated hybrid domain. The proposed method is simple yet effective and well-motivated. Extensive experiments demonstrate the outstanding performance and effectiveness of the proposed approach.

**Strengths:**

- Figure 1 is a crucial component of this paper, effectively summarizing the existing training framework and highlighting the distinctions of the proposed CMT for unsupervised domain adaptation in 3D object detection.
- This paper is well-motivated, addressing three significant challenges in this task: (1) Inadequate domain-invariant features, (2) Incomplete labels, and (3) Noisy labels. To address these issues, the author proposes three corresponding solutions: (1) mean-teacher learning with a hybrid domain, (2) a hard instance mining module, and (3) a class-aware contrastive learning module.
- The writing is well-structured and easy to follow.
- The paper presents comprehensive experiments across three adaptation tasks using three different 3D detection backbones. The superior performance against eight baselines demonstrates the effectiveness of the proposed method in adapting to different 3D scene domains.

**Limitations:**

- The primary limitation of this paper is its exclusive focus on the car category, rendering the proposed method inapplicable to real-world scenarios where multiple categories (e.g., pedestrians, cyclists) coexist. ReDB [1] introduced a multi-class domain adaptation framework for 3D object detection, training and adapting a single detection model for three different object categories simultaneously. It is essential to compare the CMT proposed in this paper with ReDB in a multi-class setting to demonstrate its effectiveness in adapting to more realistic 3D environments.
- Some typo in writing. 1) In line 113, all “figure” should be consistent as “Figure”.

[1] Chen, Z., Luo, Y., Wang, Z., Baktashmotlagh, M., & Huang, Z. (2023). Revisiting Domain-Adaptive 3D Object Detection by Reliable, Diverse and Class-balanced Pseudo-Labeling. In Proceedings of the IEEE/CVF International Conference on Computer Vision (pp. 3714-3726).

**Clarify of Initial Rating:**
I am inclined to weakly accept this article due to its comprehensive experiments, outstanding results, simple and effective methods, and high-quality writing. However, the focus on a single object category does not fully represent a realistic 3D scenario. If the rebuttal does not include multi-class experimental results, I would need to lower my rating.

**Suitability:**

2

---

### Official Review · Reviewer_bWWG · 2024-05-25

**Rating:** 3
**Confidence:** 4

**Summary:**

The authors introduce the Co-training Mean-Teacher (CMT) framework for unsupervised domain adaptation in 3D object detection, utilizing both source and target data to create a hybrid domain that enhances feature alignment incorporating hard instance mining, class-aware contrastive learning, and batch adaptive normalization.

**Strengths:**

(1) The paper is well-formulated with delicate figures and tables. The writing is easy to follow and understand.

(2) Extensive experiments demonstrate the effectiveness of the proposed methods with a significant margin.

**Limitations:**

(1) The paper focuses solely on unimodal 3D object detection tasks, which does not align with the multidisciplinary scope of ACM MM.

(2) The proposed framework primarily merges existing Co-Training and Mean-Teacher methodologies, raising concerns about its foundational motivation and novelty.

(3) For a paper on unsupervised domain adaptation (UDA), the authors fail to provide empirical or mathematical evidence to demonstrate that the domain gap has indeed been narrowed.

**Suitability:**

1

---

### Meta-Review · Area_Chair_dn8b · 2024-07-01

**Recommendation:** Accept (Poster)
**Confidence:** 4

**Metareview:**

This study focuses on unsupervised domain adaptation for 3D object detection, aiming to adapt 3D detectors across different 3D scene domains. This paper proposes a co-training mean-teacher framework that incorporates hard instance mining and training with a generated hybrid domain. The proposed method is simple yet effective and well-motivated. Experiments demonstrated the outstanding performance and effectiveness of the proposed approach.

After the rebuttal, all reviewers are positive about the submission.